# OMEGAlpes, an Open-Source Optimisation Model Generation Tool to Support Energy Stakeholders at District Scale

Sacha Hodencq [1,*], Mathieu Brugeron [1], Jaume Fitó [2], Lou Morriet [1], Benoit Delinchant [1] and Frédéric Wurtz [1]

1    University Grenoble Alpes, CNRS, Grenoble INP, G2Elab, F-38000 Grenoble, France; mathieu.brugeron@g2elab.grenoble-inp.fr (M.B.); lou.morriet@g2elab.grenoble-inp.fr (L.M.); benoit.delinchant@g2elab.grenoble-inp.fr (B.D.); frederic.wurtz@g2elab.grenoble-inp.fr (F.W.)
2    Laboratoire Optimisation de la Conception et Ingénierie de l'Environnement (LOCIE), CNRS UMR 5271—Université Savoie Mont Blanc, Polytech Annecy-Chambéry, Campus Scientifique, Savoie Technolac, CEDEX, 73376 Le Bourget-Du-Lac, France; eng.fito@gmail.com
*    Correspondence: sacha.hodencq@g2elab.grenoble-inp.fr

**Abstract:** Energy modelling is key in order to face the challenges of energy transition. There is a wide variety of modelling tools, depending on their purpose or study phase. This article summarises their main characteristics and highlights ones that are relevant when it comes to the preliminary design of energy studies at district scale. It introduces OMEGAlpes, a multi-carrier energy modelling tool to support stakeholders in the preliminary design of district-scale energy systems. OMEGAlpes is a Mixed-Integer Linear Programming (MILP) model generation tool for optimisation. It aims at making energy models accessible and understandable through its open-source development and the integration of energy stakeholders and their areas of responsibility into the models. A library of use cases developed with OMEGAlpes is presented and enables the presentation of past, current, and future development with the tool, opening the way for future developments and collaborations.

**Keywords:** energy modelling; MILP optimisation; district scale; open-source; stakeholders; use cases

## 1. Introduction

The current reality of global warming invites us to rethink and reshape our human activities. Energy consumption of human activities accounts for more than 70% of greenhouse gases (GHG) emissions worldwide [1]. Energy policies are now recommended with pillars such as energy sufficiency, efficiency, and the development of renewable energy sources to replace fossil fuels [2]. These worldwide challenges manifest themselves at local scales, especially in urban districts. Districts consume more than half of the global primary energy, on one hand. On the other hand, they could be the new place for energy production, with the development of low-carbon and decentralised renewable energies, as well as the increase in energy recovery potential. At such scale, energy consumption is cross-sectoral, with energy needs such as heating that can be met by electricity, gas, or direct heat use. This multi-energy contribution can come from various sources and networks. Moreover, the development of renewable energy sources brings about new challenges, such as intermittency. The design and management of multi-energy systems at district scales becomes crucial to face these challenges. Work at district scale also requires taking into account historical and new energy projects stakeholders in the transition process; supporting these actors in the process of designing energy systems that include more variable and diffuse sources is also of prime importance [3,4].

Energy modelling tools can play the role of decision support for energy stakeholders by accompanying them in learning about, sizing, and operating local energy systems. There exists a wide range of energy modelling tools with various modelling capabilities. The accessibility of these tools for energy stakeholders is dependent on their openness, their user friendliness, and their consideration of social aspects.

The aim of this article is to underline the challenges of energy modelling at district scale and to present the tool OMEGAlpes as well as its associated use cases library. OMEGAlpes is an optimisation tool developed to easily generate multi-carrier energy system models in order to support stakeholders in the first steps of system studies. The tool includes well-suited characteristics for preliminary energy system design phases, with a compromise between computational time and degree of modelling detail, as well as a semantics close to human understanding that enables optimisation based on energy, exergy, or specific stakeholders' criteria. Moreover, open-source principles have been applied in the tool development, and energy stakeholders can be integrated in the modelling. Finally, all the use cases developed with OMEGAlpes are gathered in a library and can be accessed, modified, and reused online. This allows both the models to be brought to society and society to be considered in the models.

This paper is structured as follows. First, we explore the energy modelling tools' characteristics based on literature reviews, beginning with the importance of energy system preliminary design. Then, we introduce the OMEGAlpes framework characteristics and structure. We finally present the use cases library associated with the tool that allows for the understanding, accessibility, and reproducibility of the studies.

## 2. Energy Modelling Tools Characteristics

### 2.1. Definition and Variety of Energy Modelling Tool

There is a very wide range of energy modelling tools [5–7]. As underlined by Chang et al. [7], reviews focused on various aspects of energy modelling tools, such as their characteristics and challenges with descriptive overviews; their practical applications and effectiveness; and their policy relevance and transparency. Previous literature enabled the identification of the main characteristics that should be investigated when developing or choosing an energy modelling tool. We propose a way of considering these characteristics in order of significance, from most to least discriminating. It enables the identification of a relevant energy modelling tool for a given purpose.

1. Modelling and analytical approach [6–12].
2. Space and time scale and resolution [3,6–8,12].
3. Sector coupling [3,6,7,9–11].
4. Level of accessibility and openness [5–7,11–13].
5. User friendliness and training requirements of the modelling tools [6,7,10].
6. Social aspects and policy relevance [7,12–15].

These main characteristics are discussed in the following sub-sections, with a focus on early design phases of district energy modelling. The literature also underlines other characteristics such as the technical capabilities of the modelling tools [5,9], with questions regarding flexibility features [6], model coupling, uncertainty analysis [7], computational time [3,11], and technological granularity [12]. General concept characteristics are also underlined, such as the flexible level of accuracy, the modelling language type, and the abstraction level [5,9]. Finally, some authors insist on the importance of accurate modelling of district heating [3,7,10].

### 2.2. From Study Phase to Modelling and Analytical Approach

2.2.1. Energy System Study Phases

Properly identifying the aim and the phase of the study in which an energy modelling tool is intended to be used is a first step before considering modelling approaches. Energy system design, as with any design, includes various phases that can be defined as study, concept, definition, and development [16]. Fernandez et al. [17] point out that at the very start of the design process, in a phase that we call preliminary design, there is obviously low knowledge about the system to design and therefore high uncertainties. At the same time, there is the greatest scope for action, so this is where decisions have the most impact on the study goals. This preliminary design phase includes a high degree of freedom, while being the most cost-influencing step of the design process [18], as underlined in the fields

of architecture [19], aeronautics [16], and power electronics [20]. It is therefore important to explore the widest range of possibilities with relevant modelling and decision support tools. This exploration will allow for making the right choices in the initial phases. It becomes even more relevant if integrating as early as possible aspects usually decided very late in the design cycle, such as the optimal management strategy. This is of particular importance when considering both environmental and financial costs at stake for district-scale energy systems. Thus, preliminary design will be one of the focuses of the rest of this literature review.

### 2.2.2. Modelling Approach

Energy modelling tools can first be differentiated by considering their purpose and modelling approach. Energy modelling approaches are generally separated into scenario, operation, and planning [8,9]: scenario approaches are used to determine a range of desired future situations according to a given objective such as GHG emissions reduction; operation approaches determine the technical (and potentially economic, social, or environmental) feasibility of such scenarios; and planning approaches take into account long-term evolutions [10].

### 2.2.3. Analytical Approaches

The variety of energy modelling tools brings about a diversity of analytical approaches, which can be differentiated between simulation, optimisation, and equilibrium models [7]; power systems and electricity market models; and qualitative and mixed methods scenarios [21]. Setting macroeconomic approaches aside, we focus on simulation and optimisation models. Simulation models are often used to approximate a physical system's actual behaviour and evolution [9]. The operation of the system is imposed by a given set of parameters. Simulation models are quite common, including, for instance, dynamic thermal simulation used for energy modelling in buildings [22,23] and the use of tools such as Citysim [24] or the DIMOSIM platform [25]. Optimisation models describe the physical system as an optimisation problem, that is to say, as optimisation variables subject to constraints and aiming for a given objective function to minimise. The general purpose of optimisation models is to determine the operation and/or the sizing of an energy system. A multitude of potential solutions to the optimisation problem are explored, with respect to the constraints and objective.

When it comes to choosing the analytical approach in preliminary design, one should consider that the model parameters will be highly uncertain. Generally speaking, the more complex the model, the more precise its characteristics will have to be, and therefore the greater its uncertainty. As illustrated in Figure 1, potential errors in the performance of the system will therefore result from both the lack of accuracy of the model and the data uncertainties, which increase with the complexity of the model [26]. As a result, macroscopic models whose physical complexity is not too high should be favoured in preliminary design. At the same time, the models should be capable of handling the variety of possible solutions due to the large number of degrees of freedom, in order to translate the functioning of the system while limiting the parameterisation of the model and therefore its total uncertainty.

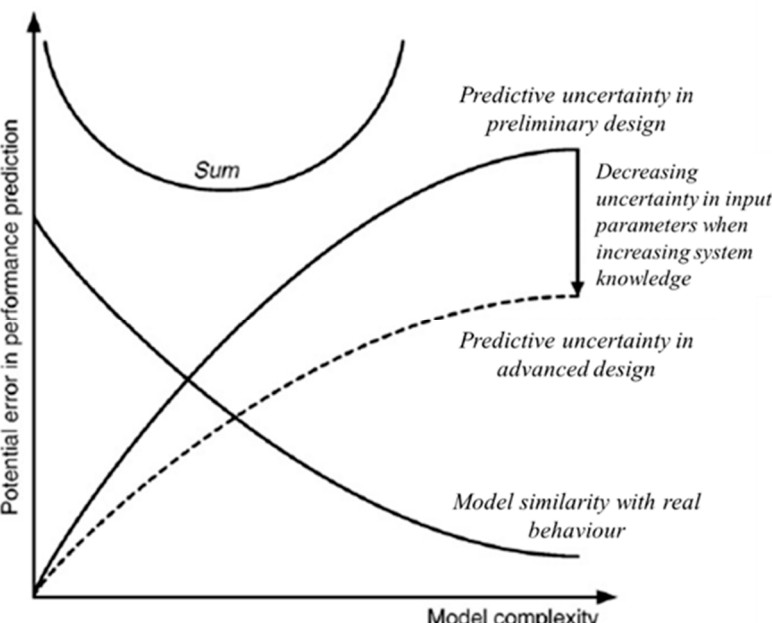

**Figure 1.** Model uncertainty vs. complexity in design phases. Source: authors from Trčka et al. [26].

Simulation is by nature based on the definition of study scenarios. This implies potentially time-consuming negotiations between the different solutions studied and a dependency on a priori determined management instructions without any degree of freedom. Simulation often forces repeated trial and error resolution [27], while optimisation enables exploring a wide range of decision variables at the same time. Optimisation, then, seems more suitable for goal-seeking within complex systems [11,28], especially in early design phases where a lot of studied scenarios are still possible. Special attention should be paid to ensuring democratic process with optimisation models: Lund et al. [29] underline that if optimisation and simulation modelling approaches can actually be used together, simulation is well-suited for democratic decision-making, while optimisation risks imposing a unique expert solution.

Various optimisation methods can be used in energy modelling. These include heuristic methods with genetic algorithms and particle swarm optimisation; stochastic optimisation; and distributed optimisation for consensus problem using game theory. Linear and quadratic programming can also be employed, as well as Mixed-Integer Linear Programming (MILP) [30,31]. MILP combines several interests compared to other optimisation methods. First, many systems, such as storages, present finite states that binaries better enable expressing compared to Linear Programming (LP) [6,8]. During the preliminary design, a lot of variables are not set yet, which entails a lot of optimisation variables especially at high temporal resolution. At the same time, the modelling includes linear flows, such as energy and finance. MILP optimisation can lead to the global optimum more quickly than other methods for such energy optimisation problems [9,31]. However, MILP may require data pre-processing such as piecewise linearisation, depending on the modelling detail of the technical system representation [32]. LP and MILP are used in a variety of frameworks. Respective examples include URBS [33], MODEST [34], and MESSAGE [35] for LP; and the Ehub Modeling Tool [36] and DER Cam [37] for MILP.

### 2.3. Space and Time Scale and Resolution

2.3.1. Geographical Scale

Most of existing energy modelling tools focus on large energy systems [6], allowing for wide-scale energy scenarios. This includes tools such as Balmorel [38], TIMES [39,40], and PyPSA [41]. At such regional or continental scales, technical specifications are often aggregated and do not represent individual plants or energy system components [7]. In

contrast, district-scale energy systems bring about new challenges with the development of local energy sources, such as energy systems stability and flexibility strategies on both the production and demand sides [8]. Platforms such as Reopt [42] or Artelys Crystal Energy Planner [43] offer MILP optimisation modelling at local scales.

### 2.3.2. Temporal Resolution and Horizon

In the context of energy transition, capturing the actual dynamics of the modelled system is essential [7]. High temporal resolutions enable taking into account renewable energy sources as well as load variability. Large time horizons are also necessary to include both daily and seasonal energy dependencies. These temporal accuracies are especially relevant for energy systems, including their storage and flexibility strategies. However, a compromise has to be found between high temporal accuracy and acceptable computational time. Period reduction techniques from heuristic approaches, iterative approaches, and grouping algorithms [44] can be used in order to abide by this compromise. Temporal resolution and horizon also depend on the modelling approach: while single year consideration can be sufficient for operational analysis, multi-year outlooks will be needed for planning approaches [45].

### *2.4. Sector Coupling*

Traditionally, energy models only represent a single energy carrier, especially power systems [6], without considering sector coupling [11]. This reflects the current situation of energy systems that operate according to a very sectoral logic. Each energy carrier is associated with a specific energy chain, with its own networks and actors that are not very interconnected. However, cross-sector integration is thought to be essential to modelling viable sustainable energy systems, including higher shares of renewable energy sources [46,47]. Cross-sector interactions bring about synergies that can offer new flexibility and storage potentials, through conversions such as power to heat or power to gas. One could note that sector coupling goes from considering multi-carrier energy systems to integrating mobility issues or even to integrating diet and waste management.

### *2.5. Open Energy Modelling*

While historical and current mainstream approaches in energy modelling are proprietary solutions and lack transparency, open energy modelling is a promising emergence [48]. A growing number of energy modellers put into practice open-source principles and gather in communities such as the open energy modelling initiative [49,50], whose wiki identifies open models [51]. Various levels of openness can be differentiated, from transparency with datasets and model documentation available, to open development with available code and data as well as open-source development principles [52]. "Open" here refers to models, code, and data that can be freely accessed, used, modified, and shared by anyone for any purpose [53]. While closed models prevent easy review [54], model comparison [13,15], or third-party contributions [27], open energy modelling has many advantages.

First, the need for adaptation and extension to new contexts entailed by the fight against climate change is facilitated by open-source: specific features can easily be added and discussed transparently [5,12,55]. Collaboration is improved between energy stakeholders that can share both method and quantitative work [9,52,56], thus limiting parallel efforts. The models are accessible with lowered barriers for adoption relative to conventional proprietary solutions [55], both in terms of finance and access to knowledge. Then, the outcomes of energy systems studies are used to shape energy policies and affect the public. Thus, transparency and openness are needed for the affected stakeholders to participate in the decision-making process [14] and have recently been requested by institutions such as the European Union [57] or the UNESCO [58]. Furthermore, they can provide public reliability to scientific expertise in the field and enable public engagement [59], which has been identified as key in the success of energy transition projects [60]. Finally, open energy modelling practices enable improving scientific quality through natural exchange

of knowledge and eased peer-review, thus limiting errors [56], biases [52], or even fraud. The open energy modelling tools are highly maintainable [55] as well as continuously improved [5]. With respect to proprietary software, their source code remains available over time, and they can meet high standards [9]. As Oberle et al. point out, such tools are "on the right road to achieving a competing level of accuracy, while also providing a much higher level of transparency" [12].

Open energy modelling faces challenges such as lack of awareness and practical knowledge on these issues, lock-in to proprietary software, and institutions' inertia [48]. In the academic world, open-source principles of early and regular release can go against usual practices. Some may fear for their reputation or for the time they spend on support, even if experience suggests collaboration interests outweigh this issue [52]. Organisations can also tend to withhold information and ideas, in order to create their own unique and closed models and to receive certain funding. Open energy modelling would need further evolution in policy and scientific practices in order to develop and become the norm.

### 2.6. User Friendliness and Accessibility

Another issue that comes with open energy modelling is information overload, identified by Cao et al. [61]: making all the code and data available may make the models very tough to understand and use. User friendliness then becomes another desirable characteristic of energy modelling tools, indicating whether the tool can be easily handled and understood, for instance, through a graphical interface [6,10,62]. Documentation to associate with versioned open-source code helps in understanding the tool, as well as tutorials and beginner examples, in addition to execution environments. Hülk et al. [63] also offer factsheets to present frameworks, models, and scenarios in a clear, concise, and understandable way. Means of synchronous or asynchronous socialisation (e.g., forums), if used by a sufficiently broad community, make it possible to capture and disseminate both explicit and tacit knowledge [64]. Ferrari et al., identify six user-friendly tools, among which three are free to use and none are open-source [62].

Making models semantically close to the energy stakeholders' one is also a way to make the understanding of energy modelling tools accessible. To do so, the choice of the level of abstraction is key. Low level of abstraction will be the closest to physical electronic circuits: it will consist of machine language with binaries. It can be noted that the higher the level of abstraction, the easier the energy model will be to formalise for a user and to understand for someone who would retrieve his work. High-level language, such as object-oriented programming, will make the models understandable and facilitate their use [5]. In particular, using Python enables taking advantage of a modern scripting language including object-oriented capabilities as well as a wide range of functions widely used in energy modelling [9]. It also facilitates model identification, extraction, and external use [48]. Model-based formulation goes further, differentiating the model used by the modellers and stakeholders from the one used by the solver algorithm, and simplifying the formulation of complex problems by coming closer to human understanding [65]. However, the translation from high to machine abstraction level alone might be insufficient, especially for energy non-experts. Libraries of prebuilt models with high-level language can be developed. The model assembling is then simplified, since the modeller can use prebuilt bricks, instead of developing the whole system. Such practices are developed in model generation tools and enable capitalising on and reusing studies formulations. The library can be completed internally or externally, in order to answer the stakeholders' needs. The model capitalisation is then simplified by object-oriented programming inheritance, assembling new bricks from existing ones with specific functionalities. For instance, a lead-acid battery brick can inherit from a generic storage brick. Figure 2 illustrates these abstraction levels and their accessibility to various stakeholders. Here we differentiate three kinds of stakeholders: actor who is involved in the energy system design, user of the energy modelling tool, and developer of the tool. A stakeholder can cumulate various roles. Actors can also become users or developers, depending on the accessibility and

user-friendliness of the energy modelling tool. The terms actors and stakeholders will be indifferently used in the rest of the article.

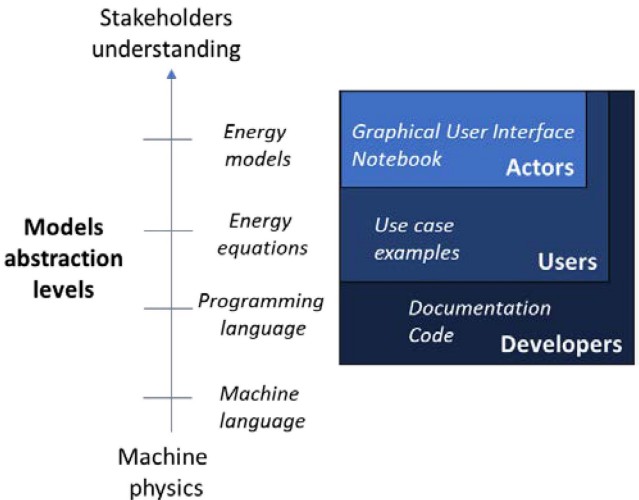

**Figure 2.** Abstraction levels of models with respect to stakeholders' understanding.

These practices are of particular interest for MILP problems formulation. However, this facilitation of formalisation can lead to a biased understanding of the modelling by the user, who will not necessarily be aware of the content of the models. Moreover, the higher the semantic level of formulation, the more transcriptions will be necessary to present the problem to the machine solver.

### 2.7. Including Social Aspects

Including social aspects in energy modelling tools has been identified as one of the current challenges of the domain [21], even if socio-technical considerations become more developed [7]. On the one hand, social aspects can be considered through the modelling of social issues, as offered by the QTDIAN toolbox from the European SENTINEL project [66]. On the other hand, social aspects can be integrated through the involvement of stakeholders in the design process. Some authors argue that this involvement is necessary, considering the impact energy systems design has on society [14,29]. Such an involvement could improve the models' relevancy and legitimacy [7]. Robbie Morrison defends the idea of an open system analysis community [50], including civil society organisations and public authorities in order to select future views through energy scenarios specifications, as well as an open energy modelling community to provide model frameworks and workflow support.

The modelling of social aspects such as policy constraints [15] or stakeholders' objectives and area of responsibility can help negotiation phases in a multi-stakeholder design process [67]. Since energy modelling and policy making influence each other, transdisciplinary models and practices as well as transparency are needed to develop common and useful tools for society [68].

### 2.8. Energy Modelling Tool Choice for Preliminary Design at District Scale

Considering this literature overview regarding energy modelling tools' characteristics, several points of interest can be underlined when it comes to the preliminary design of an energy system at district scale. Macroscopic optimisation models should be opted for in order to assess a variety of variables while keeping uncertainties low. MILP optimisation model generation tools seem relevant for such a task. High temporal resolution and multi-carrier energy systems should be features of the tool. The energy modelling tool should abide by open-source principles, thus benefitting from their numerous advantages as well as making the tool accessible to energy stakeholders. The involvement of these

stakeholders can be fostered by the tool's user-friendliness, with an abstraction level close to human understanding, as well as by its documentation, graphical interface, and support. Finally, the modelling of social constraints and objectives represents a relevant additional feature in order to bridge the gap between energy modelling and policy making. These characteristics are presented in Figure 3 in a significance order: each choice at one level will close possibilities afterwards and will allow the tools to be differentiated from one another. The levels in this figure correspond to the previous sub-sections.

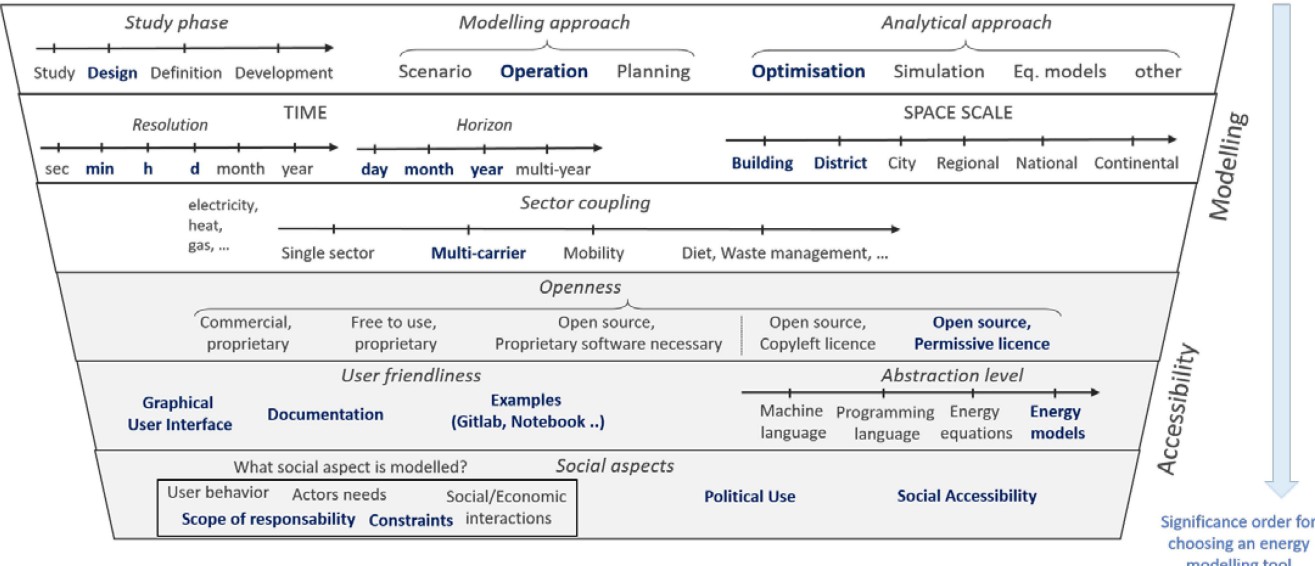

**Figure 3.** Energy modelling tool characteristics, with OMEGAlpes characteristics in bold blue. Source: authors.

Considering these characteristics, several modelling frameworks can be contemplated for the preliminary design of an energy system. OSeMOSYS [69] and Energy Scope TD [11] are open-source modelling tools using MILP optimisation and integrating sector coupling, and they respectively study long-term wide-scale projects and urban- to national-scale systems. At district scale, oemof [5], ficus [70], and rivus [71] can be used. Rivus is well-adapted to optimise the costs of distribution networks, while ficus is developed for capacity-expansion planning and unit commitment of a factory's distributed energy supply system. Oemof and its internal library solph offer an open-source model generator as well as a generic data model. Efforts are made on model coupling and transparent and collaborative development. Ficus, rivus, and oemof use the GNU-GPL 3.0 copyleft licence, while OSeMOSYS, Calliope [72] and Energy Scope are developed with the Apache 2.0 permissive licence. While copyleft licences contribute to open-source practices dissemination, permissive licences can enable collaborations with a wider range of stakeholders. If some of the presented energy modelling tools develop user-friendly features and graphical interface, none currently include social aspects in the modelling.

We present OMEGAlpes, an open-source MILP generator that supports energy stakeholders in the preliminary design of district-scale energy systems. OMEGAlpes includes the various identified characteristics and invites the energy stakeholders into the modelling process. It does this firstly through a fully transparent and open energy modelling process; secondly, through its semantic and prebuilt models close to human understanding and graphical representation; and finally, by modelling actors' constraints, objectives, and areas of responsibility. As a result, the optimisation models can become negotiation objects, and their solutions are not unique and definitive but subject to discussion among the energy project modellers and stakeholders.

### 3. OMEGAlpes, an Open-Source Optimisation Tool for District-Scale Energy Stakeholders

*3.1. General Description and Main Characteristics*

OMEGAlpes stands for Optimisation ModEls Generation for Energy Systems. Following up on the previously identified energy modelling tool characteristics, OMEGAlpes features are presented hereinafter:

1. OMEGAlpes modelling is intended to support stakeholders in the preliminary design of energy systems. It is a MILP optimisation model generation tool used for energy system operation and focuses on macroscopic representation of energy balances in order to not increase the potential error in performance prediction (see Section 2.2.3).
2. OMEGAlpes explores building- to district-scale energy systems. Time steps are generally hours but can be adjusted from minutes to days. The time horizon can be adjusted from days to years.
3. OMEGAlpes is a multi-carrier energy modelling tool. The two mainly used energy vectors are electricity and heat, but gas is also usable.
4. OMEGAlpes is developed in the freely available scripting language Python with the Apache 2.0 licence [73], a permissive open-source licence only requiring attribution to the source code authors. OMEGAlpes source code is available on a Gitlab repository [74] and its documentation on the readthedocs website [75]. OMEGAlpes leaves the choice of solver to the user through the PuLP package [76]: the use of open-source solvers does not limit the accessibility to professionals or academic users [7], while the possibility of using proprietary solvers can greatly improve the tool performance, as open-source solvers are often outperformed by commercial ones [77]. A factsheet presenting OMEGAlpes is available in the Open Energy Platform factsheets library [78].
5. The semantics of OMEGAlpes is thought to be close to that of the energy stakeholders, with pre-built energy units representing consumption, production, storage units, or assembled units such as conversion or reversible units. Operational constraints and objectives are available in these units. They were developed both in collaboration with stakeholders on actual projects and to explore research questions, and they are continuously improved and completed. A graphical representation is associated with the energy units and enabled developing the first version of a graphical interface enabling users to directly generate an OMEGAlpes script. In addition to its online documentation, OMEGAlpes provides a use cases library (named *omegalpes_examples* in the Gitlab repository), including beginner examples as well as article study cases, and provides Jupyter Notebooks (hereinafter referred to simply as notebooks) in order to easily grasp the tool modelling principles and functionalities. These notebooks can directly be used and modified online via the Mybinder [79] public service. This use cases library is detailed in Section 4.2.
6. OMEGAlpes includes an actor package, modelling the energy stakeholders' objectives, constraints, and responsibility scope. The energy and actors' models can be quickly adapted to the stakeholders' needs. This, in addition to its openness and abstraction level, makes its models relevant for negotiation support. Currently, OMEGAlpes is indirectly used for policy support in the manner of Chang et al. [7]: models are used for negotiation and advice in a waste heat recovery project with a local laboratory involved in the modelling, as well as the local public authority and district heating network operator. The French Agency for ecological transition (ADEME) has also assessed the OMEGAlpes tool, its methods, and its use cases.

Regarding its other general characteristics, OMEGAlpes includes unit and integration tests in order to ensure OMEGAlpes source code behaves as intended, which is particularly relevant for collaborative development.

OMEGAlpes is a decision support tool for the design of district-scale and multi-carrier energy systems. It couples MILP optimisation general capabilities with energy and exergy modelling. It also includes the stakeholders (which we also call actors) in the preliminary design studies. To do so, the tool provides an energy system representation close to the

actors' understanding, with a range of functionalities adapted to the field technical reality. These functionalities are fully accessible and usable thanks to open-source development features, as well as a capitalisation process that is continuously improved and completed based on studies. Moreover, it enables considering the actors' constraints and areas of responsibility. Thus, models can be used as negotiation supports between stakeholder, users and developers. A presentation of OMEGAlpes general structure is showed in Figure 4. A complete OMEGAlpes UML (Unified Modelling Language) diagram describing the tool structure is available in Appendix A. Details regarding the energy package, including the exergy module as well as the actor package, are provided in the sections hereinafter.

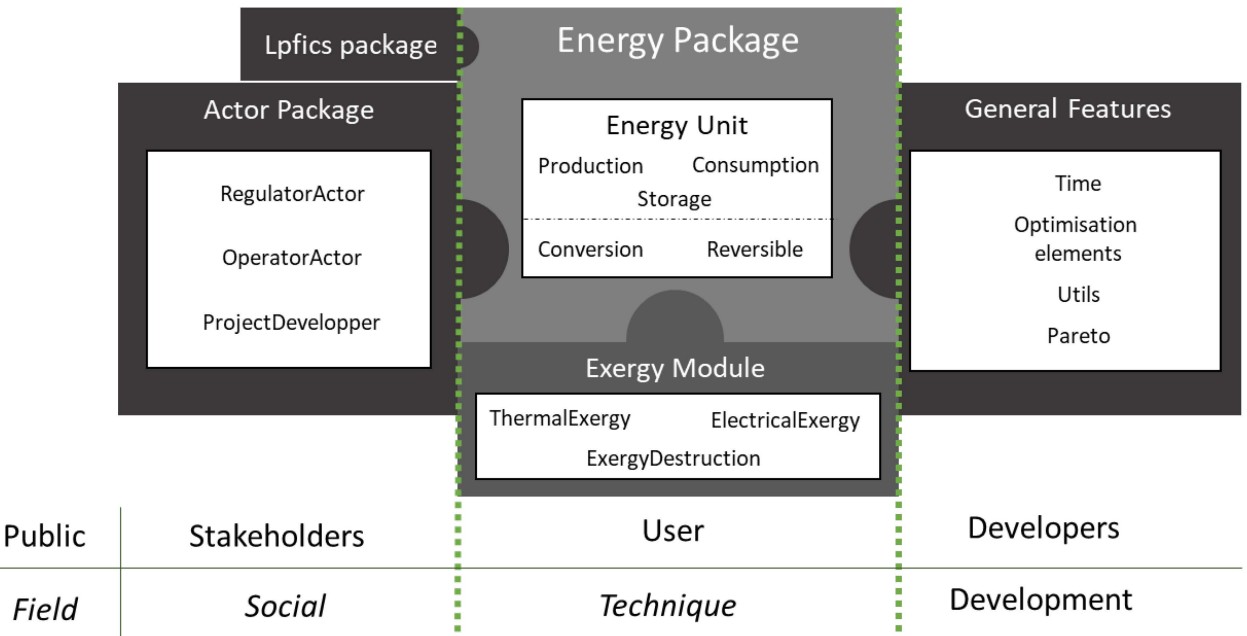

**Figure 4.** OMEGAlpes principle diagram including public and field. Source: authors.

### 3.2. Energy Modelling in OMEGAlpes

#### 3.2.1. Energy Library

In this section, we focus only on the technical aspects of energy optimisation at district scale by describing the process from the optimisation problem's definition to its mathematical solving. Here, the aim is to represent the energy flows balance by modelling the energy system, related specific constraints, and associated objectives that translate the study goals. The OMEGAlpes modelling process is based on a macroscopic approach considering the energy flows exchanged between the different elements of the energy system. The energy model is developed using object-oriented programming based on the principle of technical bricks. Thus, the code associated with an element is grouped within the same *EnergyUnit* class as the classes that inherit from it. They are divided into five kind of energy units:

- Production unit (*ProductionUnit*), to model, for example, a windmill or a solar power plant.
- Consumption unit (*ConsumptionUnit*), to model, for example, a dwelling or a domestic electrical appliance.
- Storage unit (*StorageUnit*), to model, for example, a solar water tank or batteries.
- Reversible unit (*ReversibleUnit*), to model energy systems which can non-simultaneously produce and consume energy, such as electrical machines.
- Conversion unit (*ConversionUnit*), from one energy vector to another, for example a heat pump.

*ProductionUnit*, *ConsumptionUnit* and *StorageUnit* are considered as mono-energy vectors and can support a unique energy source from heat, electricity, or gas. Their power-flow type and direction are defined with the object *EPole*. One should note that the terms "production" and "consumption" are used here with regards to considered study scope and sector. For example, the windmill could be considered as a conversion unit from wind mechanical energy to electricity. *ReversibleUnit* and *ConversionUnit* can support multiple energy vectors (for example, thermal to electric). This representation allows the user to integrate each of the elements constitutive of an energy system. Various kinds of *EnergyUnit* class are modelled and include their own set of constraints and available objective functions. This enables the user to specify an energy unit type and related input parameters depending on his needs. For example, *FixedEnergyUnit* type needs an energy profile as an input, and *ShiftableEnergyUnit* type needs a time profile that can be shifted on a given horizon. Constraints and objectives also inherit from dedicated classes of the general package. Constraint types will be detailed in the actor package sub-section (see Section 3.3.2). Once again, please refer to Appendix A for a comprehensive vision of OMEGAlpes structure.

Energy units are linked together using energy nodes (*EnergyNode*). Each energy node must ensure energy balance between the units connected to it, as well as the connection of a single energy carrier. Thus, the energy optimisation problem at the district scale is built on two levels: at the first level, by modelling the energy system through a set of energy units representative of its constituent elements; and at the second one, by ascertaining the energy balance through intermediary energy nodes. The energy model is only an abstract overlay to assist the user but does not allow the problem to be solved, i.e., optimised. To do so, it is necessary to translate the model for the solver to run it. The PuLP package enables this translation [76]. The chosen solver then gives the results of the optimisation problem, which are then provided to the user in the pot-processing phase. In addition to the energy-modelling possibilities that have been presented, the tool provides additional options for technical evaluation through the exergy library, described hereinafter.

### 3.2.2. Exergy Library

Energy analysis alone can be insufficient to assess the quality of energy streams in a system or to offer optimal ways to transform energy [80]. The exergy module serves the primary purpose of determining exergy destruction within a unit. The package uses three main classes as pillars: '*ElectricalExergy*', '*ThermalExergy*', and '*ExergyDestruction*'. The first two identify, respectively, the electrical or thermal flows interacting with an energy unit, and they determine the corresponding exergy flow accordingly. Then, the "ExergyDestruction" module reads the exergy flows assessed for the unit and applies an exergy balance to determine its exergy destruction.

- The module '*ElectricalExergy*' determines the exergy flow associated with an input, output, or accumulation of electricity in a consumption, production, or storage unit, respectively.
- The module '*ThermalExergy*' determines the exergy flow associated with an input, output, or accumulation of heat in a consumption, production, or storage unit, respectively.
- The module named '*ExergyDestruction*' allows determining exergy destruction within a unit and enabling that magnitude as an optimisation objective. This module requires an exergy efficiency ('*exergy_eff*') and the dead state temperature ('*temp_ref*') to be specified for exergy analysis. In the case of thermal storage units, their temperature level ('*temp_heat*') is also required. The dead state temperature ('*temp_ref*') defaults to 20 °C, as in the case for the "ThermalExergy" module. Users can change this value, but must be aware that all units in the study must have the same '*temp_ref*', for the sake of thermodynamic consistency. The module also requires that the exergy flow of the unit have been determined pre-emptively, by the means of either "ElectricalExergy" or "ThermalExergy".

Users should be aware of a few methodological simplifications within the *exergy* module. Firstly, it is not intended for assessing the following forms of exergy: chemical, potential, and kinetic. A second simplification concerns the assessment of thermal exergy. Namely, the exergy of transferred heat depends strongly on temperature levels. Despite that, the *exergy* module assesses thermal exergy assuming that temperature levels remain constant. Lastly, users should be aware that the term "*ExergyDestruction*" assessed within the module aggregates two concepts: losses of energy (and thus of exergy) from the unit to its surroundings and thermodynamic irreversibility within the unit itself. Experts in exergy analysis typically call the former 'exergy losses', the latter 'exergy destruction', and 'irreversibility' the aggregate thereof. Regardless, the *exergy* module assesses only the aggregate. This simplification is acceptable; energy planners focus mainly on overall efficiency, and do not need the details on the sources of irreversibility. The exergy library documentation describes the module thoroughly, by detailing the choices for each specific *EnergyUnit*.

### 3.3. Actor Package

#### 3.3.1. Contributions of the Actor Logic to the Energy Modelling Layer

OMEGAlpes offers a library of models associated with the "multi-actor" approach through the actor package [67]. An explicit actor model aims to differentiate technical issues from stakeholder-induced issues and to differentiate the decisions made by the stakeholders. This model does not attempt to model the actors as such but is restricted to the constraints and objectives carried by the actors that influence the project framework. This is a matter of being able to differentiate the technical part of the model from the part associated with the actors. When modelling, this approach can lead stakeholders to question what is technical and what is related to actor's choices. It also helps to provide additional insights in stakeholder negotiations by identifying whether the issues are technical or actor-related. Negotiation can then be facilitated by the fact that decisions are associated with their stakeholders, and their influence on the project can be assessed. The modelling logic behind the actor package is debatable, especially as, by refining the technical considerations, it is possible to take it into account on the energy layer. However, we are convinced that bringing stakeholders into the modelling loop can facilitate the technical refinement of the energy model and can help decision-making and negotiations between stakeholders.

#### 3.3.2. Actor Classes, Constraints Typing, and Related Objectives

The Actor class is defined as an abstract class, i.e., not specific to a particular actor, and has generic methods for adding or removing constraints and goals. Regarding the constraints, a distinction is made between:

- definition constraints (*DefinitionConstraint*), used to calculate quantities or to represent physical phenomena (e.g., storage state of charge) considered a priori as non-negotiable;
- technical constraints (*TechnicalConstraint*), used to represent technical issues (e.g., operating power of a production unit) considered a priori as non-negotiable unless technical changes are made;
- actor-specific constraints, *ActorConstraint* and *ActorDynamicConstraint*, used to model constraints that are due to actor decisions (e.g., minimal level of energy consumption over a period).

Those constraints are exclusive, and only actor-specific constraints are a priori negotiable as decided by the stakeholders themselves. Actors modelling includes additional objectives. In OMEGAlpes, a weight can be added to an objective in order to give more importance to one or more objectives compared to the others.

#### 3.3.3. Actor Categories

Three typical categories of actors have been identified: regulators, operators, and developers. The first formulate the rules and procedures, while the other two must respect them. This distinction was used to define the tree structure of the actor package.

- Regulators do not operate any particular energy unit but influence the energy system with regard to grid and/or resource regulation. Their decisions can affect all energy units.
- Operators can only influence—with respect to constraints and objectives—the units within their area of responsibility, as defined in the following sub-section. Based on a typology of operator actors, we have developed the following classes: *Consumer*, *Producer*, *Prosumer*, and *Supplier*.
- Developers are modelled by the *ProjectDeveloper* class. It is derived from the *Actor* class in order to add objectives and constraints that are specific to the actors carrying out the project.

### 3.3.4. The Principle of the Actors' Area of Responsibility

In order to link the energy units to the actors, an area of responsibility is defined for each actor in the model. These are the energy units they operate (for the operating stakeholders) or build (for the developers). From a modelling point of view, this notion of area of responsibility limits the constraints and objectives of the stakeholders, which can only be applied to the area of responsibility of the latter.

### *3.4. OMEGAlpes Features for Meta-Analysis of the Energy System Optimisation Problem*
### 3.4.1. Algorithms for Identifying Incompatible Constraints: lpfics

A method for identifying constraints when the problem is infeasible has been developed. This method is implemented in the lpfics package (i.e., linear problems: find infeasible constraint sets [81]). This package relies on constraints typing in two ways. The first one directly gives the infeasible constraint set and thus offers an additional degree of negotiation according to the types of the identified constraints, as the constraints' names, types, and formulations are provided. The other approach is to identify the constraints according to their typology. For this purpose, the constraints are separated upstream into sub-groups according to their type.

### 3.4.2. Calculation and Representation of Pareto Fronts

Pareto fronts can be used in negotiation processes to assess two antagonistic objectives. OMEGAlpes integrates a feature that calculates and plots Pareto fronts. The use of MILP modelling requires the formulation of single objectives to be provided to MILP solvers. As a result, obtaining Pareto fronts requires using the weighting method, known as an effective approach [82].

### 3.4.3. Graphical Representation and Interface

We offer a graphical representation to show the model of the technical system. We differentiate the energy units in several ways. A first differentiation is the energy vector: yellow for gas, red for heat, blue for cold, and purple for electricity. A second differentiation is the type of energy unit, represented by a symbol: a triangle with a fixed bar below the triangle if the consumption is fixed or an arrow if consumption is variable. A third differentiation refines the energy unit type with the place and direction of this symbol: towards the unit for energy consumption, outwards for production, two butterfly-shaped triangles for storage, etc. Arrows represent the power flows, and the energy nodes are represented by circles. It is also important to represent the constraints and objectives associated with the technical system being modelled: they are shown in rectangles. Figure 5 presents a sample of the graphical representation. The complete graphical representation convention is available in OMEGAlpes documentation [75].

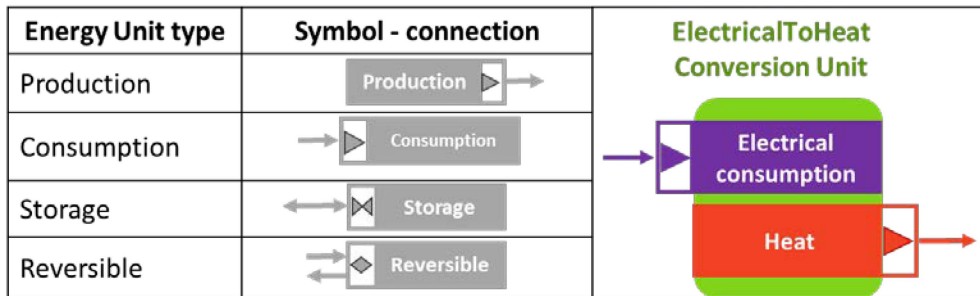

**Figure 5.** Sample of OMEGAlpes graphical representation.

A graphical interface facilitates access to the modelling to allow the model to be represented as a medium for exchange between stakeholders. Thus, the OMEGAlpes-Web graphical interface has been developed to provide a modelling tool. The graphical interface allows a user to generate a model from the OMEGAlpes model library, first graphically and then in Python code. It is based on the graphical representation elements presented above [83]. For the moment, only a small part of OMEGAlpes has been developed, mainly the parts of the energy package presented hereinbefore. Moreover, it is necessary to launch the python file by oneself to obtain an optimised solution.

## 4. Presentation of OMEGAlpes Functionalities: The Example of Waste Heat Recovery

*4.1. Description of the Energy System of the Waste Heat Recovery Project Example*

In order to present OMEGAlpes functionalities and principles in an intuitive way, we have chosen a case study that is simple with regards to the energy aspects of the system, but which may include a certain technical and social complexity. The objective of this case study is both to show the detailed definition of an energy model for a district energy system and to put into perspective OMEGAlpes' additional capabilities.

The case study consists of the recovery of waste heat from an electro-intensive industrial actor. Waste heat, also called excess heat, can be defined as dissipated heat that is not the purpose of a given system [84]. Waste heat recovery interests and challenges are discussed in Section 5.2.1. This example enables presenting a relevant use of the exergy and actors' functionalities. The energy units can be separated into three zones:

- The first zone contains the industrial actor (EII).
- The second zone contains the district heating network operator (DHNO).
- The third zone contains the distribution electricity network operator (DENO).

Table 1 presents the different energy units engaged in this use case.

**Table 1.** Presentation of the energy units in the use case and their related zones.

| Industrial | Thermal Supplier | Electricity Supplier | Recovery System |
|---|---|---|---|
| Heat production ($EII_{out}$)<br>Electricity consumption ($EII_{in}$)<br>Heat dissipation (DISS) | Heat production ($DHN_{in}$)<br>Heat consumption ($DHN_{out}$) | Electricity production ($DEN_{in}$)<br>Electricity consumption ($DEN_{out}$) | Thermal storage (TS)<br>Heat pump (HP) |

In the industrial zone, heat production represents the waste heat dissipated in order to maintain its industrial processes, and electricity consumption represents the EII's electricity consumption resulting from its industrial activities. Then, heat dissipation represents the waste heat dissipated during the industrial processes that is not recovered and therefore is rejected into a river.

In the district heating network zone, heat production represents all the thermal power plants connected to the district heating network, and heat consumption represents the district heat consumption.

In the electricity supplier zone, electricity production represents the electricity imports coming from the power grid, and electricity consumption represents the electricity exports provided to the EII and the heat pump from the grid.

The waste heat recovery system is constituted by two energy units. The heat pump enables supplying heat from the industrial waste heat to the district heating network by increasing the waste heat temperature to the network one. The thermal storage compensates for the temporal and thermal power mismatches between the waste heat and the district heat consumption.

Figure 6 presents a scheme of the use case.

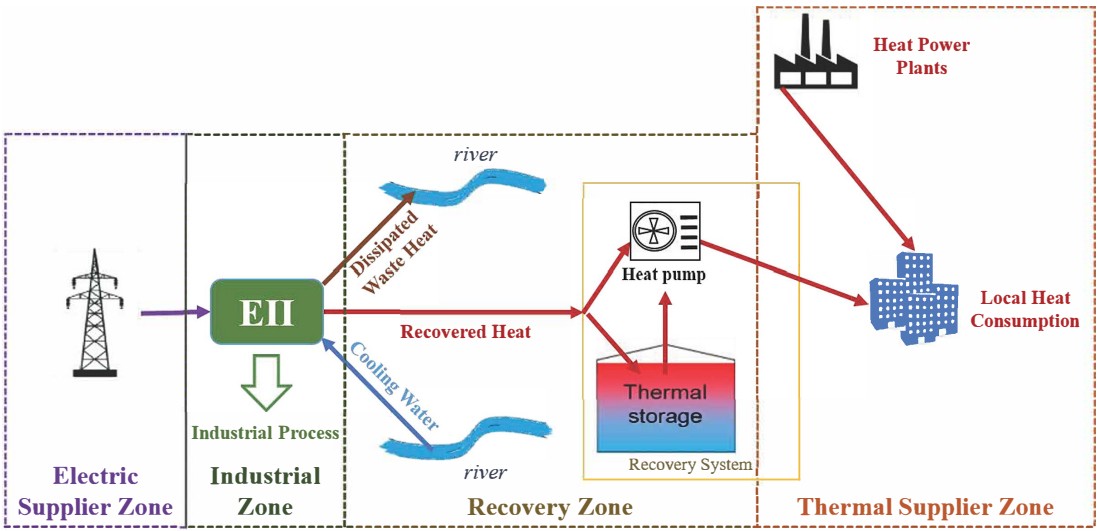

**Figure 6.** Thermal scheme of the waste heat recovery use case.

In the next sub-section, the energy model of the use case will be detailed, considering the total energy balances. Then, the use case modelling using the tool OMEGAlpes is developed. Finally, applications of the additional packages and functionalities are presented in order to highlight the related additional options

### 4.2. Waste Heat Recovery Energy Model

4.2.1. OMEGAlpes Optimisation Process Modelling

OMEGAlpes energy modelling for the use case of waste heat recovery is performed considering several simplifying hypotheses and is supported by model equations. All those pieces of information, as well as detailed code and results, are available in the related notebook specifically edited for this article (see the Supplementary Materials section). The aim of this section is to present OMEGAlpes modelling principles rather than an exhaustive description of its source code.

OMEGAlpes provides to the user the opportunity to integrate all the energy balance equations by using the system of energy units. First, the modelling process consists of defining the energy units corresponding to each energy system. Table 2 shows this modelling association between energy systems and OMEGAlpes energy units.

The EII system is represented by an *ElectricaltoThermalConversionUnit* that links the EIIout waste heat of the industrial zone to the EIIin electrical consumption. The dissipation (DISS) is represented by a *VariableConsumptionUnit* that is connected to the EIIout waste heat of the industrial zone.

The DHN includes heat production and heat consumption, which are respectively represented by a *VariableProductionUnit* and a *FixedConsumptionUnit*. Indeed, HS is an adjusting variable in the heat balance and needs to be variable in order to fulfil the heat needs of the network.

ES represents the pseudo-infinite electric grid. That is why it is represented by a *VariableReversibleUnit* that can both import and export power.

HP is represented by a *HeatPump* that converts low temperature waste heat into network temperature heat. TS is represented by a *StorageUnit* whose capacity can be an optimisation variable or a fixed parameter, depending on the aim of study. In the sizing phase, capacity is considered as an optimisation variable. In the energy management phase, capacity is considered as known and provided as a parameter.

**Table 2.** Correspondence between energy systems and energy units.

| Zone | Energy System [1] | Energy Unit | Energy Type |
|---|---|---|---|
| Industrial | $EII^{in}$ $EII^{out}$ | ElectricaltoThermalConversionUnit | Electricity Heat |
| | DISS | VariableConsumptionUnit | Heat |
| District heating network | HS DHC | VariableProductionUnit FixedConsumptionUnit | Heat Heat |
| Electric supplier | ES | VariableReversibleUnit | Electricity |
| Recovery system | $HP^{in,elec}$ $HP^{in,heat}$ $HP^{out}$ | HeatPump | Heat |
| | TS | StorageUnit | Heat |

[1] With HS being the district heating network Heat Supply, DHC the District Heat Consumption, and ES the Electricity Supplier.

In order to respect the energy balances of the units that have been defined in the previous section, it is necessary to define the connection elements between the energy units. As explained earlier, this connection is made possible through the use of energy nodes, which connect a set of units to each other. Table 3 lists all the energy nodes necessary for the modelling of the case study and the associated energy units.

**Table 3.** List of energy nodes to define the energy balances to model the case study.

| Energy Node | Connected Energy Units |
|---|---|
| Electricity node | ES, $EII_{in}$, $HP_{in}$, elec |
| EII node | $EII_{out}$, DISS, |
| Recovery node | $HP_{in}$ heat, TS |
| DHN node | $HP_{out}$, HS, DHC |

Figure 7 represents the OMEGAlpes model using the OMEGAlpes graphic convention, including energy units and nodes symbols, energy carriers colours, and the main constraints and objective.

### 4.2.2. Objective Function

In order to complete the modelling of the optimisation problem representing the case study, it is necessary to define the objective functions that can be associated with it. Two main objective functions can be implemented for this case:

- Minimising the output of heat generation plants.
- Minimising the size of storage.

In the purely technical scenario presented here, only the objective of minimising production is retained corresponding to an operation study phase, but we will see later that it is possible to envisage a Pareto study that compromises between minimising storage size and minimising production.

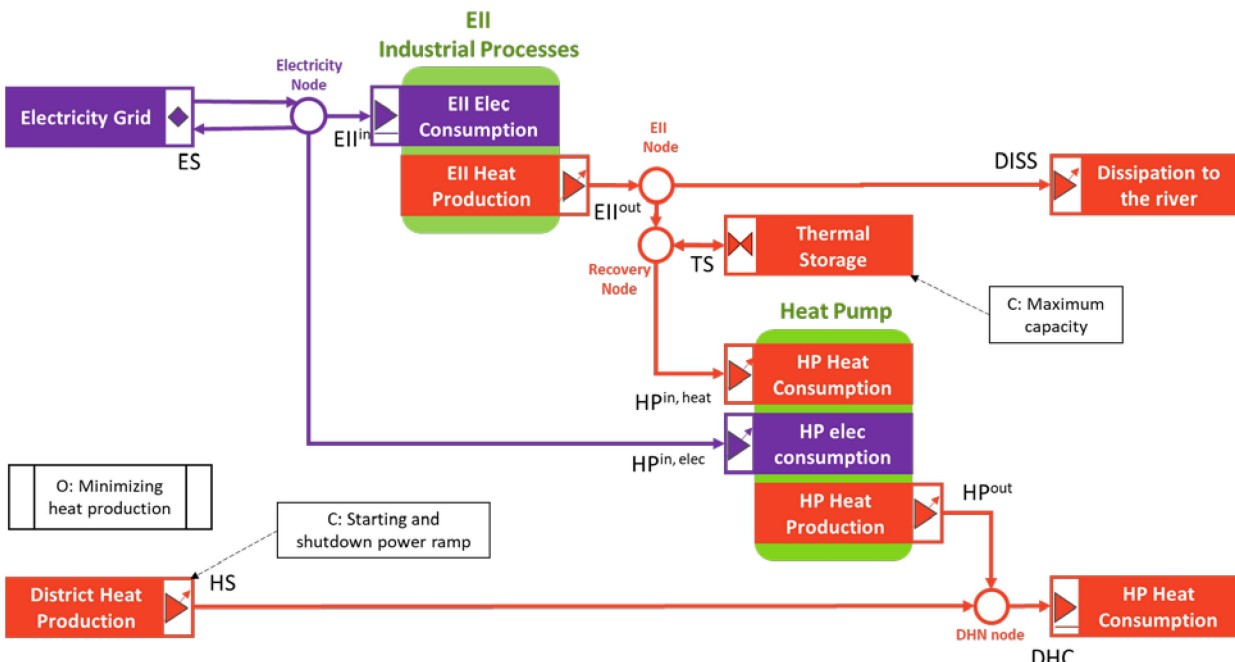

**Figure 7.** OMEGAlpes graphical representation of the waste heat recovery use case, including the main constraints and objective.

### 4.2.3. Applying OMEGAlpes Pareto, lpfics, and Exergy Functionalities to the Use Case

After focusing on the technical aspects of the use case, this sub-section describes additional features and packages. They provide more ways to study the optimisation problem (Pareto and lpfics features) or to provide more optimisation options considering both social and technical aspects (respectively, actor package and exergy module).

As introduced in the presentation of the OMEGAlpes tool, it is possible to carry out a Pareto study between two objectives, in this case minimising the heat supplier production on the network at the same time as the size of the thermal storage. The size of the storage is directly related to its capacity and so to the heat recovery potential, but it is also directly related to its capital expenditures. It is therefore possible to draw a Pareto front between these two antagonistic objectives, which can be of use for compromise and negotiation on the system design.

lpfics is a package that can be used when the optimisation problem is facing infeasibility issues. As explained in the section on lpfics (Section 3.4.1), it is possible to obtain access to a set of constraints in which there are one or several infeasible constraints. An example of its use for the waste heat recovery use case is detailed in the corresponding notebook.

The exergy module enriches the technical aspects of the energy model by implementing exergy concepts into the energy units. In the case study, this translates into the evaluation of the exergy destruction of each of the thermal units, which are the dissipation (DISS), the thermal storage (TS), the heat pump (HP), the heat generation plants (HS), and the district heat consumption (DHC). By associating a class of calculation for exergy and exergy destruction with each of them, it becomes possible to evaluate the exergy destroyed and thus to consider this evaluation as an additional objective of the optimisation problem of this case study. Such an objective often offers a unique solution with respect to energy objectives. Moreover, the economic notions that can be considered, based on the cost of energy, can be applied in the same way to exergy, thus giving an exergo-economic evaluation. This makes it possible to provide a wider range of technical assessments with energy, economic, exergy, and exergo-economic concepts [80].

### 4.2.4. Modelling Actors in the Waste Heat Recovery Use Case

Based on technical considerations, the energy modelling of the case study can lead to a certain distance from the realities of the stakeholders. For this case study, we will present scenarios highlighting the possibilities of technical understanding from stakeholders, along with the potential for negotiations and decision-making that the package allows to be highlighted. Of course, these choices are only examples among the possible actors to be modelled in this use case. As a first step, the number of actors, their types, and their areas of responsibility should be defined. Table 4 provides a description of these actors. It shows that the heat pump and thermal storage are not distributed in the areas of responsibility of the actors. This raises the question: how to enable the evaluation of the choice of the actor responsible for the recovery system? This brings about two primary scenarios, corresponding to the EII and the DHNO that are responsible for the recovery system.

**Table 4.** List of actors present in the case study with their respective areas of responsibility.

| Actors | Actor Type | Area of Responsibility | |
|--------|-----------|------------------------|---|
| EII | Prosumer | Energy Units | $EII^{in}$, $EII^{out}$, DISS |
| | | Energy Nodes | Electricity Node, EII Node |
| DHNO | Heat grid operators | Energy Units | HS, DHC |
| | | Energy Nodes | DHN Node |

In order to evaluate the two scenarios, the technical aspects of the model must be reconsidered. We now assume that the areas dedicated to thermal storage are of different sizes depending on the actor, with the DHNO having access to wider areas. This results in the consideration of a maximum thermal storage capacity that differs depending on which actor is responsible for it. Furthermore, as shown in Table 5, the placement of the storage system may differ depending on who is responsible.

**Table 5.** Modifications of the energy model depending on the scenario.

| Scenario | Maximal Thermal Capacity | Energy Node Modifications |
|----------|--------------------------|---------------------------|
| EII | $capa_{EII}^{max}$ | None |
| DHNO | $capa_{DHNO}^{max}$ | TS either linked to recovery node or DHN node |

For the case study, we propose changing the location of the thermal storage from the EII node to the DHN node. Finally, the maximum thermal storage capacity as well as the objective will be considered via actors' constraints and actors' objectives and are discussed as such.

Afterwards, we can introduce notions of regulations through a *RegulatorActor*. This allows the user to define specific actor constraints, such as a maximum energy dissipation or maximum GHG emissions threshold. This has the effect of enriching the energy model by considering regulatory aspects from the point of view of constraints and objectives.

Finally, the actor package allows to consider constraints applied by one actor to another. For example, if we refer to the notions of start-up and shutdown times of production plants, we can assume that the DHNO actor would accept heat injections in the network only if it exceeds a duration $\Delta T_{min}^{inj}$, considering the start-up times and a safety margin. This constraint is then added at the EII's heat injection level and can widen the scope for negotiation. Indeed, there are cases in which this constraint leads to the non-injection of heat production whose duration is close to the duration $\Delta T_{min}^{inj}$. From this point of view, it becomes possible to bring about additional negotiation elements and to consider a re-evaluation of the minimum injection duration.

With these different examples, we wanted to show to what extent it is possible to enrich the technical aspects of the model with this actor approach. We remind the reader that this list is not exhaustive, but that it is intended to highlight a range of situations that are representative of the options possible with the actor package.

## 5. Use Cases Library

### 5.1. Use Cases Library Description and Interests

The development of OMEGAlpes has been based on actual and theoretical case studies, which have allowed the enrichment of the library of energy models and development of new functionalities. This section introduces OMEGAlpes' use cases library, presenting the history of the development of the tool and the areas of interest. The capitalisation of use cases makes it possible to keep track of the whole history of the analysis, describing not only the results but also how they were obtained [64]. While the articles provide an understanding of the use cases, the notebooks allow users to go further. The notebooks can be accessed and used at any time by any reader. The notebooks include code as well as details that cannot be presented in the linked article, such as the origin of the data, the modelling methods, etc. Notebooks can serve as collaborative objects thanks to their intermediate level of complexity compared to the framework source code or complete documentation. In addition, other notebooks are available to serve as tutorials for the use of the tool. The objective is to provide a library of examples as educational as possible in order to provide a maximum understanding of the modelling possibilities of OMEGAlpes.

The case studies have been classified according to two fields: waste heat recovery and energy autonomy, mainly consisting of PV self-consumption. Table 6 presents these case studies with their associated publications and specifications. In the actors column, LNCMI stands for French National Laboratory for High Magnetic Fields, which is an electro-intensive laboratory. The CCIAG is the Grenoble district heating network operator.

**Table 6.** OMEGAlpes use cases library including article references and languages, source code, notebooks, space and time scale and resolution, developments, and research objective.

| Use Case | Article Reference | Source Code [1] | Notebook [2] | Actors | Space and Time | Functionalities Development | Research Objective |
|---|---|---|---|---|---|---|---|
| Waste heat recovery | [85] EN | UCE1 | NB1 | LNMCI CCIAG | District scale Year study Hourly time steps | Prototype of OMEGAlpes Energy package—differentiation of energy vectors | Development of energy planning decisions for an electro-intensive consumer, subject to economic, social, and environmental constraints and objectives |
| | [86] EN | | | LNMCI CCIAG | District scale 6 months study Hourly time steps | Shiftable energy unit | Methodology of data-driven modelling of an existing load profile in order to build archetypes |
| | [8] EN | | | LNMCI CCIAG | District scale Two weeks study 10 min time step | Heat pump class Thermal Modelling of buildings | Evaluation of energy flexibility for building heating |
| | [87] FR | | | LNMCI CCIAG | District scale One year study | - | Technical, economic, and environmental consequences of the evolution of research infrastructure, from component to district scale |
| | [80] EN | | | LNMCI CCIAG | District scale One year study Hourly time step | Exergy module | 4E (energy, exergy, economy, and exergo-economy) methodology description and evaluation |
| | [88] EN | | | LNMCI CCIAG | District scale One year study Hourly time step | Exergy module | Evaluation of different technical criteria and their possible impact on the design of a waste heat recovery system |
| | [89] EN | | | LNMCI CCIAG | District scale One year study Weekly resolution | Exergy module, energy planning modification | Evaluation of flexibility and temperature management strategies to improve waste heat recovery energy- and exergy-wise |
| | [90] FR | | | LNMCI CCIAG | District scale One year study 7 weeks planning | Exergy module, actor package | Multi-objective optimisation for the design of a waste heat recovery system by means of multi-actor technical and economic analysis |
| | [91] FR | | | LNMCI CCIAG | District scale 2 weeks horizon, hourly resolution | Open science considerations—LNCMI notebook | Describing and discussing open and collaborative platforms |
| | [92] EN | | | LNMCI CCIAG | - | Open energy modelling process | Presenting a transferable workflow to make open energy modelling principles and advantages accessible |

**Table 6.** *Cont.*

| Use Case | Article Reference | Source Code [1] | Notebook [2] | Actors | Space and Time | Functionalities Development | Research Objective |
|---|---|---|---|---|---|---|---|
| Energy self-consumption | [27] EN | UCE2 | NB2 | Producer Consumer Supplier Prosumer | Building scale One day study 5 min time-step | Energy and actor package | First presentation of OMEGAlpes with a simple self-consumption example |
| | [67] [EN] | UCE3 | NB3 | Prosumer Supplier | Building scale One day study Half-hourly time step | Actor package | Offering a method to help stakeholders to formalise their constraints and objectives and to negotiate in the design process |
| | FR | UCE4 | NB4 | Prosumer Consumer | Building scale One day study Half-hourly time step | Actor, lpfics | Development of a conflicting constraint identification algorithm for optimisation problems |
| | [64] EN | - | NB5 | | Building scale 4 days study Hourly time step | Open science consideration | Offering an open and reproducible use case of PV self-consumption, taking into account energy and environmental indicators |

[1] Please refer to the use case example list in the Supplementary Materials section; [2] Please refer to the notebook list in the in the Supplementary Materials section.

### 5.2. Waste Heat Recovery Use Case

#### 5.2.1. Interest of the Use Cases Field

One of the main interesting points regarding waste heat recovery lies in its positive impact on energy transitions thanks to reduction of GHG emissions. Waste heat represents an important and underexploited potential low-carbon heat source. However, waste heat recovery brings about many challenges. In the design phase, technical challenges include modelling accuracy of district heating, location, temperature management between source and consumption, and the permanence of the heat potential [93]. Waste heat recovery projects include many stakeholders such as industry, local authorities, and district heating network operators. These actors' negotiation can have a great influence on the results of the possible energy optimisation.

Regarding OMEGAlpes development, an actual waste heat recovery project is under study with the support of the stakeholders, whose presence allows for greater access to data and a better level of technical knowledge. This link with the stakeholders and the field reality enables considering their energy context, including changes in tariffs, new schedules for consumption profiles, or the study of different cooling systems and associated outlet temperatures. On the one hand, stakeholders benefit from OMEGAlpes both through technical decision support from its results and through the formalisation of their actual constraints and objectives. On the other hand, the development of the tool is enriched with new and updated pre-built units as well as functionalities coming directly from the field. As waste heat recovery involves various energy carriers, and particularly thermal power with various temperature levels, it has allowed exergy analysis to be carried out and exergy indicators to be developed. Their relevancy is tested with stakeholders. Finally, this case study can present modelling issues including uncertainties and access to energy profiles, modelling both prospective and existing units with sufficient accuracy, and the management of the storage unit.

#### 5.2.2. History of the Use Cases and Publications

Initially, this type of case study made it possible to develop the multi-vector aspects of the tool. It also allowed the energy library to be expanded, while refining the models of the energy units, such as the storage units, with the notion of charge cycle. Units such as the *HeatPump*, derived from the *ConversionUnit*, were implemented. Then, the exergy aspects were developed in collaboration with the LOCIE thermal laboratory and applied in the evaluation of the LNCMI waste heat recovery project. This collaboration continued with the study of an additional actual use case, focusing on a wider area with a variety of energy systems including solar PV, solar thermal, waste heat recovery, power grid, batteries, and heat pumps for the energy supply of a mixed district [94,95]. This collaborative development was made possible thanks to open science logic, which is an incentive for inter-laboratory collaborations and thus for envisaging enriching energy modelling tools.

### 5.3. Energy Self-Consumption Use Cases

#### 5.3.1. Interest of the Use Cases Field

As waste heat recovery use cases, energy self-consumption use cases perfectly fit into energy transitions. They enables increasing the share of renewable energies covering our energy consumption, thereby reducing the constraints on distribution networks as well as grid losses. Moreover, they can lead to energy sufficiency behaviour by making prosumers aware of their consumption. A need for comparable studies has also been identified in this field [96], while new a European regulation [97] defining citizen energy collectives has brought interest to collective self-consumption. Most of the use cases in the OMEGAlpes library are theoretical and have enabled the development and testing of OMEGAlpes functionalities. They put into perspective the negotiation possibilities offered by the OMEGAlpes tool. The actor package has been developed in collaboration with the social science laboratory PACTE, based on field studies in the domain. This type of

case study benefits from a great diversity in its implementation, with flexibility studies on consumption, battery pack sizing, or incentives at the actor level. In the preliminary studies, many sizing choices are preponderant and may depend on the actors. An actual preliminary design of an autonomous energy system was an opportunity to consider technical elements that had not been considered yet, which allowed the implementation of new classes such as *AssemblyUnit* and *ReversibleUnit*.

5.3.2. History of the Use Cases and Publications

Firstly, the various published works provided the occasion to develop the notebook library in order to allow their sharing and their reproducibility according to the logic of open science. The ONIRI project has allowed the collaboration with a French artist collective that, due to its needs, has enriched the OMEGAlpes technical library. In addition, this project allowed the development of a notebook accessible to a non-technical public. Finally, with the ORUCE project, one of its research goals was to develop open energy modelling at district scale, for instance, by offering an Open Energy Platform factsheet [98] at such scale.

**6. Outcome, Perspectives and Conclusions**

*6.1. Outcome*

In this sub-section, the authors would like to underline the main outcomes of the article, in order to highlight the article's points of interest. Through reading this article, the authors first would like the readers to consider the importance of the accessibility of energy modelling tools for democratic decision support. OMEGAlpes development and use offered a way to implement this accessibility, mainly through open-source development as well as actors' consideration through their modelling and a choice of relevant abstraction level. Readers can go further in their understanding of the tool by accessing a library of use cases, detailed for various applications and research questions. The use cases are designed to be understandable, usable, and appropriated at will. Ultimately, one can consider exchanging with the OMEGAlpes development team for collaboration, either for a use of the tool or for development of their own project.

*6.2. Perspectives*

This work includes several perspectives. First, regarding OMEGAlpes itself, improvements consistent with the reviewed literature could include firstly a better modelling of district heating, based on an actual local waste heat recovery project, and the inclusion of exergy post-treatments in the tool. These improvements could also include an update of the graphical interface in order to make it up to date with the tool's latest functionalities and enable online model solving. An economy package is also in development, in order to go further in economics considerations than those presently considered with existing economic parameters. Uncertainty calculation and management is an additional potential improvement for the tool.

Current work also focuses on the development of an open and collaborative digital platform in order to foster user interaction and collaboration based on complete open energy modelling processes in use cases. The public of this platform would range from fellow researchers to local authorities, engineering offices, and citizen collectives. This platform would enable bringing the use cases and associated energy models to society, in addition to bringing society into models through actor package development. Use cases will be further developed, including both current use cases such as photovoltaic self-consumption and waste heat recovery and new ones exploring maters of energy sufficiency and autonomy. These use cases will bring new collaborations with an engineer's office for collective self-consumption. The tool is also going to be put to good use for teaching materials in engineering and architecture schools.

Finally, on the topic of collaboration, an additional perspective on OMEGAlpes concerns its link with other energy modelling tools. This can be established through hard or

soft model coupling [Chang2021], as well as through collaboration with external teams of energy modellers. OMEGAlpes' developer team tried to develop relevant functionalities for district scale energy projects, such as actors and exergy considerations. The question of integrating such functionalities into an existing open energy modelling framework via collaboration and contribution, thus putting into practice open science principles, still needs to be explored.

### 6.3. Conclusions

This article presented OMEGAlpes, an open-source model generation tool for district-scale energy systems' preliminary design using MILP optimisation. After reviewing the main characteristics of the energy modelling tool, the article introduced the key features of OMEGAlpes as well as their use in a simplified waste heat recovery example. The range of use cases and associated research fields explored with OMEGAlpes was also presented. OMEGAlpes' open development with the source code documentation and Gitlab, as well as the use cases library, enabled collaborations with laboratories from various fields including electrical engineering, heat, and social sciences. Common work is also lead with engineering schools, engineering offices, local authorities, and energy system operators. These collaborations bring about a connection to the field reality and OMEGAlpes' continuous improvement.

With this article, we offer understanding of OMEGAlpes' method and use cases library with respect to existing energy modelling tool. From energy project actors to energy modelling tool developers, we believe OMEGAlpes' method, library, use, and development can support negotiations in energy projects at district scale. Taking into account the stakeholders in the design of energy projects could thus favour local energy transition.

**Supplementary Materials:** The following are available online at https://www.mdpi.com/article/10.3390/en14185928/s1, the source code and notebook linked to this article, as well as the use cases code and notebooks from the OMEGAlpes examples.

**Author Contributions:** Conceptualization, M.B., B.D., J.F., S.H., L.M. and F.W.; Methodology, J.F., S.H. and L.M.; Software, M.B., J.F., S.H. and L.M.; Validation, B.D., J.F., S.H., L.M. and F.W.; Formal Analysis, J.F., S.H. and L.M.; Investigation, J.F., S.H. and L.M.; Resources, B.D. and F.W.; Data Curation, J.F., S.H. and L.M.; Writing—Original Draft Preparation, M.B, S.H., J.F. and L.M.; Writing—Review and Editing, M.B. and S.H.; Visualization, M.B. and S.H.; Supervision, B.D. and F.W.; Project Administration, B.D. and F.W.; Funding Acquisition, B.D. and F.W. All authors have read and agreed to the published version of the manuscript.

**Funding:** The authors are grateful to the ADEME (the French Agency for Environment and Energy Management) for their financial support through the RETHINE project (Réseaux Electriques et THermiques InterconNEctés). This work has been partially supported by the CDP Eco-SESA receiving fund from the French National Research Agency in the framework of the "Investissements d'avenir" program (ANR-15-IDEX-02).

**Acknowledgments:** The authors first would like to thank Camille Pajot for all the work she carried out in the development of OMEGAlpes. They also want to thank the openmod community for the rewarding online and face to face exchanges.

**Conflicts of Interest:** The authors declare no conflict of interest.

# Appendix A. Comprehensive OMEGAlpes UML Diagrams

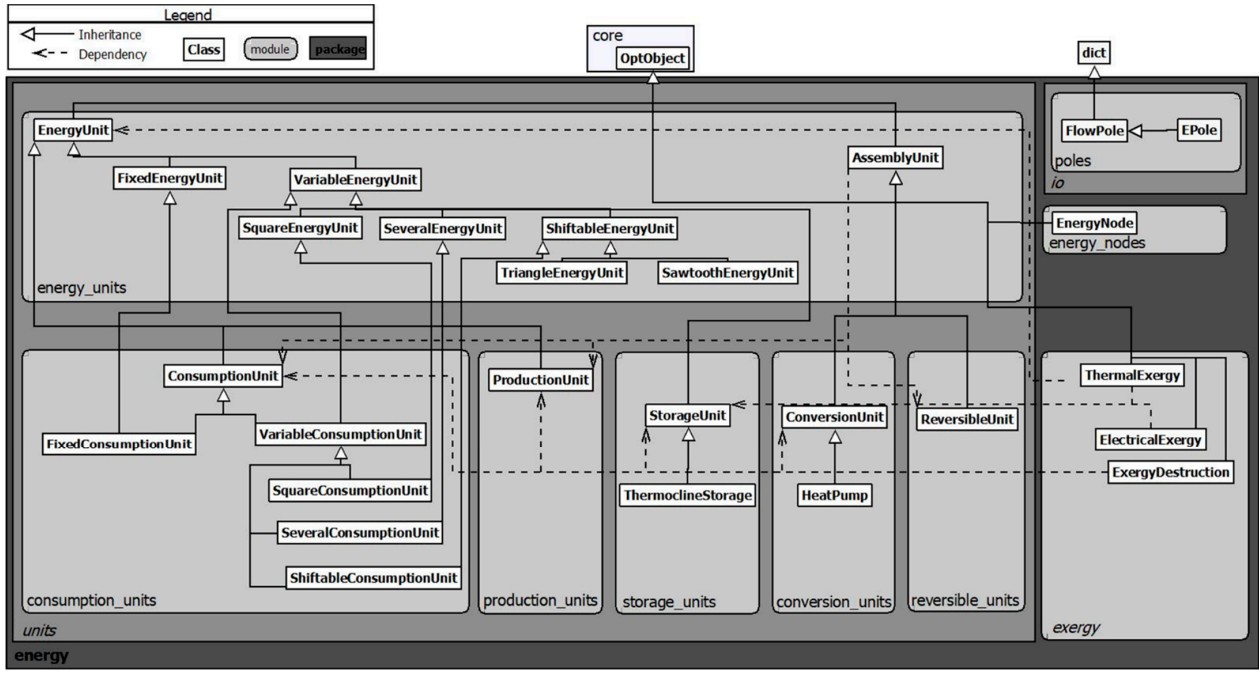

**Figure A1.** Energy package UML diagram.

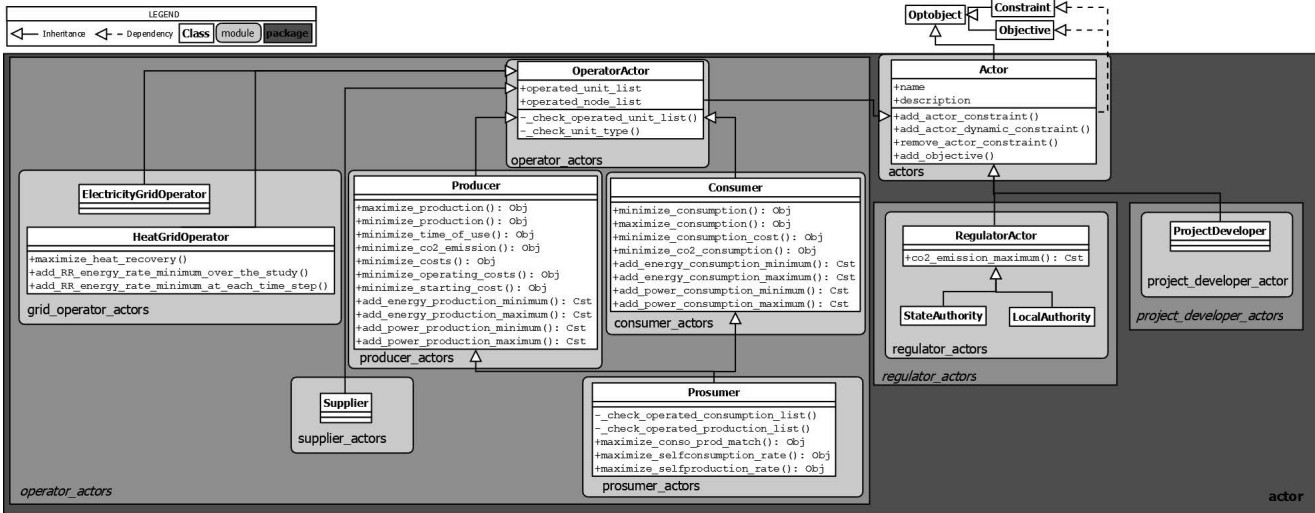

**Figure A2.** Actor package UML diagram.

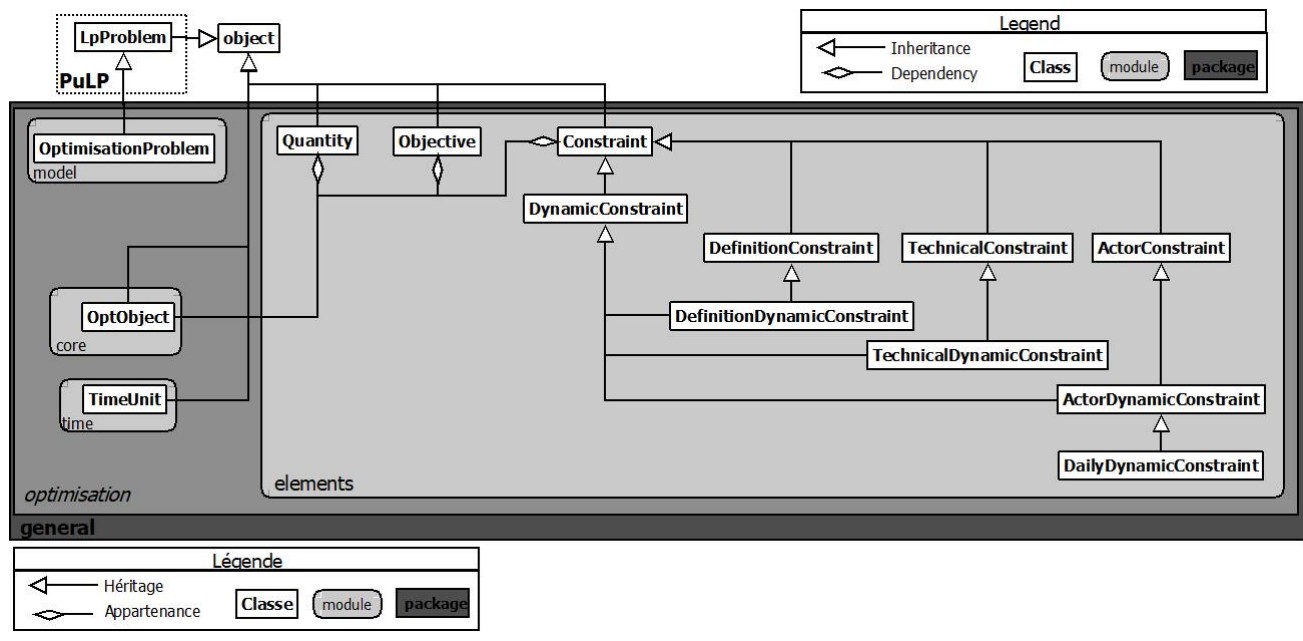

**Figure A3.** General package UML diagram.

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
