# Peer review of "OMEGAlpes, an Open-Source Optimisation Model Generation Tool to Support Energy Stakeholders at District Scale"

_energies, doi:10.3390/en14185928_

Round 1

Reviewer 1 Report

Dear authors,

many thanks for submitting your paper, it is very well written and I find no flaws in it. I recommend accepting it with out any changes.

Author Response

The authors would like to thank the reviewer for its kind judgement. The manuscript received minor changes with respect to the previous version, due to reviewers comments and clarifications or typo corrections. These changes mainly consist in a new Figure 2 in order to illustrate the level of abstraction in energy modelling, and in an outcome part (6.1.) in order to synthetise what the authors would like to transfer to the readers.

All of those changes are identified through track changes in the newly submitted version.

Reviewer 2 Report

The manuscript "OMEGAlpes, an open source optimization model generation 2 tool to support energy stakeholders at district scale" presents an important and current issue of optimizing energy models. I believe that the manuscript may be admitted to the next stages of editing, provided that the following comments are taken into account:
1. Be sure to expand the Introduction part by introducing readers to the topic of the article.
2. No clear-cut item Results, which makes it difficult to use the manuscript.
3. The Discussion point and the Conclusions in the manuscript should be distinguished separately.

Author Response

The authors would like to thank the reviewer for its relevant comments. The introduction was expanded in order to introduce readers to the rest of the article, especially accessibility issues regarding energy modelling tools. We added an Outcome part (6.1.) and distinguished it from the perspectives on one hand, and the conclusion on the other hand. The outcome part here synthetises what the authors would like to transfer to the readers. It is the result of the work carried out in the article. It should be noted that quantitative results are not directly provided in the article since the point of the article is not to underline numerical results, but such results are accessible through the notebooks.

The manuscript also received minor changes with respect to the previous version, due to reviewers comments and clarifications or typo corrections. One should note the addition of a new Figure 2 in order to illustrate the level of abstraction in energy modelling.

All of those changes are identified through track changes in the newly submitted version.

Reviewer 3 Report

All comments and suggestions for authors are contained in the attached file.

Author Response

The authors would like to thank the reviewer for his comprehensive review of the article. We updated the manuscript thanks to the various reviewers comments. Few could not be treated:

  • Line5: "Univ." is the wording the laboratory signature impose to the authors.
  • Line 1134: the typo seemed correct for the Presqu'île district in Grenoble. Moreover, it is an article title the authors no longer have access to for modification.

The manuscript received additional changes with respect to the previous version, due to reviewers comments and clarifications or typo corrections. These changes mainly consist in a new Figure 2 in order to illustrate the level of abstraction in energy modelling, and in an outcome part (6.1.) in order to synthetise what the authors would like to transfer to the readers.

All of those changes are identified through track changes in the newly submitted version.